# FusionAD: Multi-modality Fusion for Prediction and Planning Tasks of Autonomous Driving

## Abstract

Building a multi-modality multi-task neural network toward accurate and robust performance is a de-facto standard in the perception task of autonomous driving. However, leveraging such data from multiple sensors to jointly optimize the prediction and planning tasks remains largely unexplored. In this paper, we present FusionAD, to the best of our knowledge, the first unified framework that fuses the information from the two most critical sensors, camera, and LiDAR, and goes beyond the perception task. Concretely, we first build a transformer-based multi-modality fusion network to effectively produce fusion-based features. In contrast to the camera-based end-to-end method UniAD, we then establish fusion-aided modality-aware prediction and status-aware planning modules, dubbed FM-SPnP that take advantage of multi-modality features. We conduct extensive experiments on the commonly used benchmark nuScenes dataset, our FusionAD achieves state-of-the-art performance and surpasses baselines on average 15% on perception tasks like detection and tracking, 10% on occupancy prediction accuracy, reducing prediction error from 0.708 to 0.389 in ADE score and reduces the collision rate from 0.31% to only 0.12%.

## 1 Introduction

Deep Learning has been accelerating the development of Autonomous Driving (AD) in the past few years. For self-driving vehicles, the AD algorithm often takes the camera and lidar as sensory input and outputs the control command. AD tasks are usually divided into perception, prediction, and planning. In the traditional paradigm, each learning module in AD separately uses its own backbones and learns the tasks independently. Additionally, downstream tasks such as prediction and planning tasks often rely on vectorized representations from perception results, while high-level semantic information is often unavailable as in Figure 1 (a).

Previously, the end-to-end learning-based approaches often directly output the control command or trajectory based on the perspective-view camera and lidar information (Bai et al., 2022). Recent end-to-end learning approaches (Hu et al., 2023b; Casas et al., 2021; Jiang et al., 2023) start to formulate the end-to-end learning as a multi-task learning problem while outputting intermediate information along with the planned trajectories. These approaches only adopt a single input modality. On the other hand, especially through fusion with lidar and camera information for perception tasks, the perception results could be significantly improved, which has been validated in several previous work (Liu et al., 2023; Liang et al., 2022). Recently, there has been a surge of interest in BEV (Bird's Eye View) perception, particularly for vision-centric perception (Li et al., 2022b;a) as depicted in Figure 1 (b). This development has significantly advanced the capabilities of self-driving vehicles and enabled a more natural fusion of vision and lidar modalities. BEV fusion-based methods (Liu et al., 2023; Liang et al., 2022; Bai et al., 2022) have demonstrated effectiveness, particularly for perception tasks. However, the use of features from multi-modality sensors in an end-to-end manner remains unexplored in prediction and planning tasks.

To this end, we propose FusionAD, to the best of our knowledge, the first uniform BEV multi-modality based, multi-task end-to-end learning framework, with a focus on prediction and planning tasks for autonomous driving. We start from a recent popular vision-centric approach to formulate

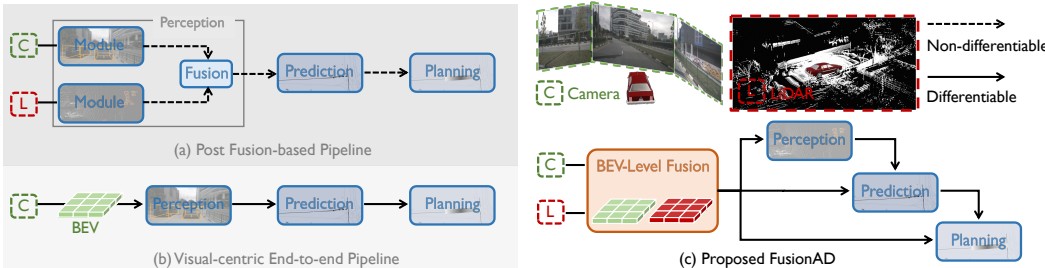

Figure 1: **Comparing different design pipelines of the autonomous driving system.** **(a)** A common practice of autonomous driving system, which consists of perception, prediction, and planning tasks. Each task is an independent task module that has its own input and output definition, and the transition between modules usually requires non-differentiable operations and prevents the system from being optimized in an end-to-end manner. **(b)** It refers to a recent end-to-end visual-centric system that learns perception, prediction and planning tasks (Hu et al., 2023b). **(c)** We present FusionAD, the first multi-modality and multi-task end-to-end learning framework that enables joint optimization of perception, prediction, and planning tasks.

our pipeline (Hu et al., 2023b). First, we design a simple yet effective transformer architecture to fuse the multi-modality information into one transformer, to produce unified features in the BEV space. As our primary focus is to explore the fusion features to enhance the prediction and planning tasks, we then formulate fusion-aided modality-aware prediction and status-aware planning modules, dubbed FMSPnP, that incorporate progressive interaction and refinement and formulate fusion-based collision loss modeling. Different from (Hu et al., 2023b), our FMSPnP module exploits a hierarchical pyramid formulation as depicted in Figure 1 (c), that ensures all tasks can benefit from the intermediate perception features. The proposed method better propagates high-level semantic information, as well as efficiently sharing features among different tasks.

We conduct extensive experiments in the popular autonomous driving benchmark nuScenes (Caesar et al., 2020) dataset. The result shows that the proposed FusionAD significantly surpasses the state-of-the-art method: a 37% error reduction for trajectory prediction, a 29% enhancement for occupancy prediction, and a 14% decrease in collision rates for planning.

The main contributions as summarized as follows:

- We propose a BEV-fusion-based, multi-sensory, multi-task, end-to-end learning approach for the main tasks in autonomous driving; the fusion-based method greatly improves the results compared to the camera-based BEV method.
- We propose the FMSPnP module that incorporates modality self-attention and refinement for the prediction task, as well as relaxed collision loss and fusion with vectorized ego information for the planning task. Experiment studies verified that FMSPnP improves the prediction and planning results.
- We conduct extensive studies in multiple tasks to validate the effectiveness of the proposed method; the experiment results show FusionAD achieves SOTA results in prediction and planning tasks while maintaining competitive results in intermediate perception tasks.

## 2 RELATED WORK

### 2.1 BEV PERCEPTION

Bird's Eye View (BEV) perception methods have gained attention in autonomous driving for perceiving the surrounding environment. Camera-based BEV methods transform multi-view camera image features into the BEV space, enabling end-to-end perception without post-processing overlapping regions. LSS (Philion & Fidler, 2020) and BEVDet (Huang et al., 2021) use image-based depth prediction to build frustums and extract image BEV features for map segmentation and 3D object detection. Building on this, BEVdet4D (Huang & Huang, 2022) and SoloFusion (Park et al., 2023) achieve temporal fusion by combining current frame BEV features with aligned historical frame BEV features. BEVFormer (Li et al., 2022b) uses spatiotemporal attention with transformers

to obtain temporally fused image BEV features. These approaches improve understanding of the dynamic environment and enhance perception results.

However, camera-based perception methods suffer from insufficient distance perception accuracy. LiDAR can offer accurate location information, but its points are sparse. To address this issue, some previous methods (Yin et al., 2021b; Bai et al., 2022) have explored the benefits of fusing multimodal data for perception. BEV is a common perspective in LiDAR-based perception algorithms (Yin et al., 2021a; Lang et al., 2019), and transforming multimodal features into the BEV space facilitates the fusion of these features. BEVFusion (Liang et al., 2022; Liu et al., 2023) concatenates image BEV features obtained by the LSS (Philion & Fidler, 2020) method with LiDAR BEV features obtained by Voxelnet (Zhou & Tuzel, 2018) to obtain fused BEV features, which improves perception performance. SuperFusion (Zhou & Tuzel, 2018) further proposes multi-stage fusion for multi-modal-based map perception.

## 2.2 MOTION FORECASTING

Following the success of VectorNet (Gao et al., 2020), mainstream motion forecasting (or trajectory prediction) methods commonly utilize HD maps and vector-based obstacle representation to predict future trajectories of agents. Building upon this foundation, LaneGCN (Liang et al., 2020a) and PAGA (Da & Zhang, 2022) enhance trajectory-map matching through refined map features, such as lane connection attributes. Furthermore, certain anchor-based methods (Zhao et al., 2020; Gu et al., 2021) sample target points near the map, enabling trajectory prediction based on these points. However, these approaches heavily rely on pre-collected High-definition maps, making them unsuitable for areas where maps are not available.

Vectorized prediction methods often lack high-level semantic information and require HD Map, thus, recent work has started to use raw sensory information for trajectory prediction. PnPNet (Liang et al., 2020b) proposes a novel tracking module that generates object tracks online from detection and exploits trajectory level features for motion forecasting, but their overall framework is based on CNN, and the motion forecasting module is relatively simple, with only single-mode output. As the transformer is applied to detection (Carion et al., 2020) and tracking (Zeng et al., 2021), VIP3D (Gu et al., 2023) successfully draws on previous work and proposes the first transformer-based joint perception-prediction framework. UniAD (Hu et al., 2023b) further incorporates more downstream tasks and proposes a planning-oriented end-to-end autonomous driving model. On the basis of our predecessors, we have carried out more refined optimization for the task of motion forecasting and introduced the refinement mechanism and mode-attention, which has greatly improved the prediction indicators.

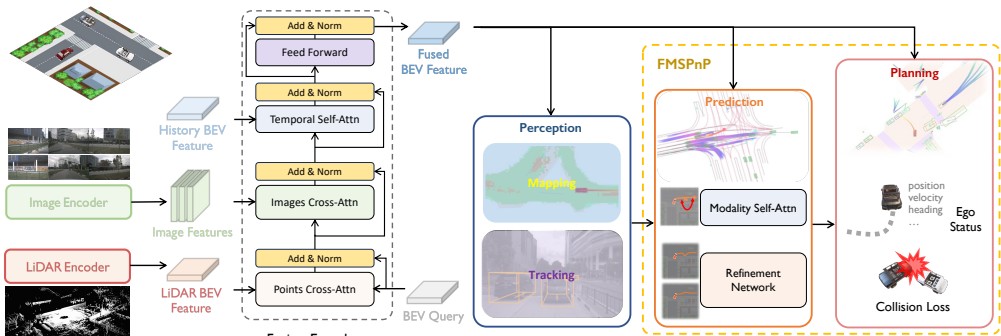

Figure 2: FusionAD Architecture Overview - FusionAD employs BEVfusion to facilitate multi-sensory, multi-task end-to-end learning specifically tailored for autonomous driving. The architecture primarily focuses on enhancing prediction and planning tasks, utilizing the fusion-aided modality-aware prediction and status-aware planning modules (FMSPnP) for these specific tasks.

## 2.3 LEARNING FOR PLANNING

Imitation Learning (IL) and Reinforcement Learning (RL) have been used for planning (Gao et al., 2022). IL and RL are used in either an end-to-end approach (Kendall et al., 2019), Chen et al. (2020)

(i.e. using image and/or lidar as input), or vectorized approach (Scheel et al., 2022; Guo et al., 2023) (i.e. using vectorized perception results as input). Even though using intermediate perception results for planning can improve the system's generalization and transparency, the vectorized approach suffers from the post-processing noise and variations of the perception results. Early end-to-end approaches such as ALVINN (Pomerleau, 1988) and PilotNet (Bojarski et al., 2016) often output the control command or trajectory directly, while lacking of intermediate results/tasks. Instead, P3 (Sadat et al., 2020), MP3 (Casas et al., 2021), UniAD (Hu et al., 2023b) learn an end-to-end learnable network that performs joint perception, prediction and planning, which can produce interpretable intermediate representation and improve the final planning performance. However, they either only make use of the lidar input (Sadat et al., 2020; Casas et al., 2021) or the camera input (Hu et al., 2023b), which limits their performance. Transfuser (Prakash et al., 2021) uses both lidar and camera input, but not in BEV space, and only performs a few AD learning tasks as auxiliary tasks. To address the issue, we propose a BEV fusion-based, unified multi-modal, multi-task framework that absorbs both the lidar and camera input.

## 3 METHOD

The overall network architecture of our proposed FusionAD is illustrated in Figure 2. Initially, the camera images are mapped to the Bird's Eye View (BEV) space using a BEVFormer-based image encoder. These are then combined with the lidar features in BEV space. Following temporal fusion, the fused BEV features are used for detection, tracking, and mapping tasks through the query-based approach. Subsequently, the tokens are forwarded to the motion and occupancy prediction tasks and planning tasks. We name our fusion aided modality-aware prediction and status-aware planning modules as FMSPnP in short.

### 3.1 BEV ENCODER AND PERCEPTION

Drawing inspiration from FusionFormer (Hu et al., 2023a), we propose a novel multi-modal temporal fusion framework for 3D object detection with a Transformer-based architecture. To improve efficiency, we adopt a recurrent temporal fusion technique that is similar to BEVFormer. Unlike FusionFormer, we use features in BEV format as input for the LiDAR branch instead of voxel features. The multi-modal temporal fusion module comprises 6 encoding layers, as illustrated in Figure 1. A group of learnable BEV queriers is first employed to fuse LiDAR features and image features using Points cross-attention and Image cross-attention, respectively. We then fuse the resulting features with historical BEV features from the previous frame via Temporal self-attention. The queries are updated by a feedforward network before being used as input for the next layer. After 6 layers of fusion encoding, the final multi-modal temporal fused BEV features are generated for the subsequent tasks. See appendix for more details.

### 3.2 PREDICTION

Benefiting from the more informative BEV features, the prediction module receives more stable and fine-grained information. Based on this, in order to further capture the multi-modal distribution and improve prediction accuracy, we introduce modality self-attention and refinement net. Details of the design can be found in Figure 3.

**Context-Informed Mode attention.** In UniAD (Hu et al., 2023b), dataset-level statistical anchors are used to assist multimodal trajectory learning and inter-anchor self-attention is applied to enhance the quality of the anchors. However, since these anchors do not consider historical state and map information, their contribution to multimodal learning is limited. Therefore, we are considering adding this operation later. After the motion query retrieves all scene context to capture agent-agent, agent-map, and agent-goal point information, We then introduce mode self-attention to enable mutual visibility between the various modes, leading to better quality and diversity.

$$Q_{mode} = \text{MHSA}(Q_u) \tag{1}$$

where MHSA denotes multi-head self-attention. $Q_u$ represents the query that obtains the context information.

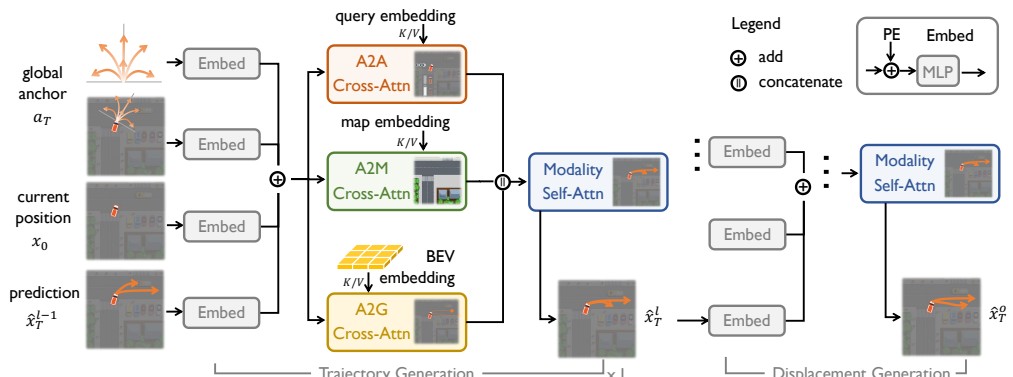

Figure 3: **Details of the prediction module in FMSPnP**. It includes two stages: *Trajectory Generation* and *Displacement Generation*. In the trajectory generation, we apply Agent-to-Agent (A2A), Agent-to-Map (A2M), and Agent-to-Goal (A2G) attention to fuse the surrounding agents, lane, and goal information. We also propose the modality self-attention to improve mode quality and diversity. In the displacement generation, we use the prediction of the first stage as the anchor input and output the offset of the current prediction. Note that the two stages share a similar backbone structure.

**Refinement Network.** Deformable attention uses statistical anchors as reference trajectories to interact with Bev features. As mentioned earlier, this reference trajectory increases the difficulty of subsequent learning due to the need for specific scene information. We introduce a refinement net to use the trajectories generated by Motionformer as more accurate spatial priors, query the scene context, and predict the offset between the ground truth trajectory and the prior trajectory at this stage. As shown below:

$$Q_R = \text{DefAttn}(\text{Anchor}_p, \widehat{x}_m, B) \tag{2}$$

where $\text{Anchor}_p$ represents the spatial prior. A simple MLP will be used to encode the trajectory output by Motionformer and perform maxpool in the time dimension to get $\text{Anchor}_p$. $\widehat{x}_m$ represents the endpoint of the Motionformer output trajectory.

Table 1: Main results for multi-tasks, end-to-end learning; $^*$ denotes evaluation using checkpoints from official implementation.

| | Detection | | Tracking | | Mapping | | Prediction | | | | Occupancy | | | | Planning | | |
|---|---|---|---|---|---|---|---|---|---|---|---|---|---|---|---|---|---|
| | mAP % ↑ | NDS ↑ | AMOTA % ↑ | AMOTP % ↓ | IoU-L % ↑ | IoU-R % ↑ | ADE $m$ ↓ | FDE $m$ ↓ | MR % ↓ | EPA % ↑ | VPQ-n % ↑ | VPQ-f % ↑ | IoU-n % ↑ | IoU-f % ↑ | DE $m$ ↓ | CR$_{\text{avg}}$ % ↓ | CR$_{\text{traj}}$ % ↓ |
| UniAD | 0.382 | 0.499 | 0.359 | 1.320 | 0.313 | 0.691 | 0.708 | 1.025 | 0.151 | 0.456 | 54.7 | 33.5 | 63.4 | 40.2 | 1.03 | 0.31 | 1.46 |
| **Ours** | **0.574** | **0.646** | **0.501** | **1.065** | **0.367** | **0.731** | **0.389** | **0.615** | **0.084** | **0.620** | **64.7** | **50.2** | **70.4** | **51.0** | **0.81** | **0.12** | **0.37** |

## 3.3 PLANNING

During the evaluation process, we do not have access to high-definition (HD) maps or pre-defined routes. Therefore, we rely on learnable command embeddings to represent navigation signals (including turning left, turning right, and keeping forward) to guide the direction. To obtain the surrounding embedding, we input the plan query, which consists of the ego-query and command embedding, into bird's-eye-view (BEV) features. We then fuse this with the ego vehicle's embedding, which is processed by a MLP network, to obtain the state embedding. This state embedding is then decoded into the future waypoint $\hat{\tau}$.

To ensure safety, during training, we incorporate a differentiable relaxation of the collision loss as (Suo et al., 2021), in addition to the naive imitation L2 loss. We present the complete design in Figure 4.

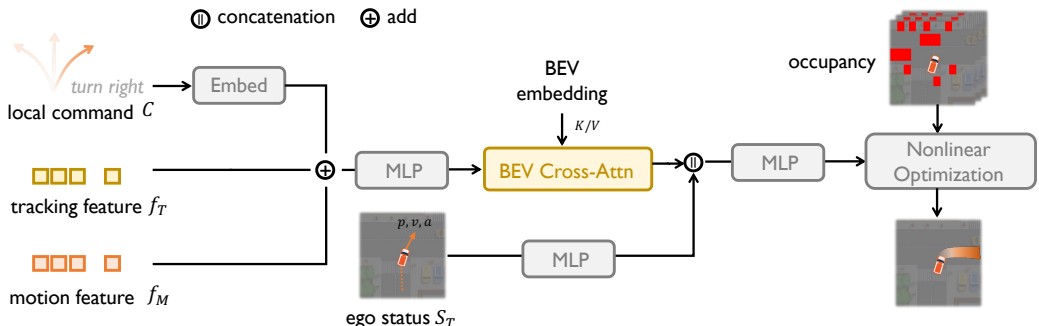

Figure 4: **Details of the planning module in FMSPnP**. We first fuse the local information with the BEV embedding to obtain the current state. Then, we enhance the current state by injecting ego information including the current agent's position, velocity, acceleration, and heading. Lastly, we optimize the planned trajectory with Newton's method to ensure trajectory safety and smoothness. We also incorporate a relaxed collision loss during the training to improve the training stability.

$$\mathcal{L}_{\text{tra}} = \lambda_{\text{col}}\mathcal{L}_{\text{col}}(\hat{\tau}, b) + \lambda_{\text{imi}}\mathcal{L}_{\text{imi}}(\hat{\tau}, \widetilde{\tau}) \tag{3}$$

where $\lambda_{\text{imi}} = 1$, $\lambda_{\text{col}} = 2.5$, $\hat{\tau}$ is the original planning results, $\widetilde{\tau}$ denotes the planning labels,and $b$ indicates agents forecasted in the scene. The collision loss is calculated by:

$$\mathcal{L}_{\text{col}}(\hat{\tau}, b) = \frac{1}{N^2}\sum_{i=0}^{N}\max\left(1, \sum_{t=0}^{P}\mathcal{L}_{\text{pair}}\left(\hat{\tau}^t, b_i^t\right)\right)$$
$$\mathcal{L}_{\text{pair}}\left(\hat{\tau}^t, b_i^t\right) = \begin{cases} 1 - \frac{d}{r_i + r_j}, & \text{if } d \le r_i + r_j \\ 0, & \text{otherwise} \end{cases} \tag{4}$$

Besides, during inference, to further ensure safety and smoothness of the trajectory, we perform trajectory optimization using Newton's method (Hu et al., 2023b) using occupancy prediction results from the occupancy prediction model.

### 3.4 TRAINING

We utilize three-stage training for the multi-sensor, multi-task learning. For the first stage, we only train the BEV encoder and perception tasks; for the second stage, we fix the BEV encoder and train the perception, prediction, and planning tasks; while for an optional third stage, we further train the occupancy and planning tasks, with fixing all other components.

## 4 EXPERIMENTS

### 4.1 EXPERIMENT SETUP

We conduct all our experiments on the A100 GPU cluster, utilizing 32 A100 GPUs for the experiment training. We use the nuScenes dataset (Caesar et al., 2020), comprising 1000 driving scenes captured in both Boston and Singapore. Each scene spans approximately 20 seconds, and nuScenes offers a vast collection of 1.4 million 3D bounding boxes encompassing 23 distinct categories, sampled at 2Hz. For our work, we use of the available camera, lidar, and canbus data. For the hyperparameters, we use $0.075 \times 0.075 \times 0.2m$ for lidar pointcloud; we use the resolution of $1600 \times 900$ for an image; the BEV size is $200 \times 200$; we use AdamW optimizer with the start learning rate of $2e - 4$, warm-up of 1000 iteration and CosineAnnealing scheduling is used; the batch size is 1 due to the high GPU memory consumption; the queue size is 5 for stage one and 3 for stage two and three.

Table 2: **The results of motion forecasting.** FusionAD remarkably outperforms previous methods.

| Method | minADE $\downarrow$ | minFDE $\downarrow$ | MR $\downarrow$ | EPA $\uparrow$ |
|---|---|---|---|---|
| PnPNet (Liang et al., 2020b) | 1.15 | 1.95 | 0.226 | 0.222 |
| VIP3D (Gu et al., 2023) | 2.05 | 2.84 | 0.246 | 0.226 |
| UniAD (Hu et al., 2023b) | 0.71 | 1.02 | 0.151 | 0.456 |
| **FusionAD** | **0.388** | **0.617** | **0.086** | **0.626** |

Table 3: **The results of occupancy prediction.** FusionAD remarkably outperforms previous methods on all metrics. "n." and "f." indicates near $(30 \times 30m)$ and far $(100 \times 100m)$ evaluation ranges respectively.

| Method | IoU-n $\uparrow$ | IoU-f $\uparrow$ | VPQ-n $\uparrow$ | VPQ-f $\uparrow$ |
|---|---|---|---|---|
| FIERY (Hu et al., 2021a) | 59.4 | 36.7 | 50.2 | 29.9 |
| StretchBEV (Akan & Güney, 2022) | 55.5 | 37.1 | 46.0 | 29.0 |
| ST-P3 (Hu et al., 2022) | - | 38.9 | - | 32.1 |
| BEVerse (Zhang et al., 2022) | 61.4 | 40.9 | 54.3 | 36.1 |
| PowerBEV (Li et al., 2023) | 62.5 | 39.3 | 55.5 | 33.8 |
| UniAD (Hu et al., 2023b) | 63.4 | 40.2 | 54.7 | 33.5 |
| **FusionAD** | **71.2** | **51.5** | **65.5** | **51.1** |

We follow (Hu et al., 2023b) to evaluate the performance of end-to-end autonomous driving tasks. Specifically, for the metrics of perception tasks, we use mAP and NDS to evaluate the detection task, AMOTA and AMOTP to evaluate the tracking task, IoU to evaluate the mapping task.

To evaluate the prediction and planning tasks, we use commonly used metrics, such as End-to-end Prediction Accuracy (EPA), Average Displacement Error (ADE), Final Displacement Error (FDE), and Miss Rate (MR) to evaluate the performance of motion prediction. For future occupancy prediction, we use the metrics Future Video Panoptic Quality (VPQ) and IoU for near $(30 \times 30m)$ and far $(100 \times 100m)$ range, adopted from FIERY (Hu et al., 2021a). And we adopt Displacement Error (DE) and Collision Rate (CR) to evaluate the planning performance, where the collision rate is considered as the main metrics.

### 4.2 EXPERIMENT RESULTS

The main experimental results are shown in Table 1. We can see that our design of fusing camera and Lidar sensory information significantly improves the performance of almost all tasks, compared to the UniAD baseline (Hu et al., 2023b). Note that we do not include any data augmentation methods, which are commonly used for perception tasks.

The motion forecasting results are shown in Table 2. FusionAD significantly outperforms existing methods. For the future occupancy prediction, we also observed that FusionAD performs much better than existing methods, especially for IoU-f and VPQ-f in $(100 \times 100m)$ range, as shown in Table 3, this indicates the fusion of lidar information is helpful for longer range.

Table 4 presents the planning results, demonstrating FusionAD's superior performance compared to existing methods, as indicated by its lowest average and total collision rates. $CR_{traj}$ denote the collision rate among whole 3-second trajectory, while $CR_{avg}$ adopted from (Hu et al., 2023b) denotes the average collision rate of trajectory at 1,2 and 3 seconds. Furthermore, FusionAD achieves the second lowest L2 distance, which serves as a reference metric to assess the similarity between the planned trajectory and the ground truth. It is important to note that the collision rate is the primary metric (Guo et al., 2023; Caesar et al., 2021), whereas in real-world scenarios, multiple viable trajectories may exist, making the L2 distance a secondary consideration.

### 4.3 ABLATION STUDIES

The ablation studies pertaining to the FMSPnP module are presented in Tables 5 and 6. Upon examination, it becomes evident that the refinement net and mode attention module significantly contribute

Table 4: Planning Results: FusionAD achieves the state-of-the-art performance in the most critical metrics, average collision rate and trajectory collision rate, surpassing both planning only methods (Jiang et al., 2023) as well as end-to-end method (Hu et al., 2023b)

| Method | $DE_{avg}$ | $CR_{1s}$ | $CR_{2s}$ | $CR_{3s}$ | $CR_{avg}$ | $CR_{traj}$ |
|---|---|---|---|---|---|---|
| FF (Hu et al., 2021b) | 1.43 | 0.06 | 0.17 | 1.07 | 0.43 | - |
| EO (Khurana et al., 2022) | 1.60 | 0.04 | 0.09 | 0.88 | 0.33 | - |
| ST-P3 (Hu et al., 2022) | 2.11 | 0.23 | 0.62 | 1.27 | 0.71 | - |
| VAD (Jiang et al., 2023) | **0.37** | 0.07 | 0.10 | **0.24** | 0.14 | - |
| UniAD (Hu et al., 2023b) | 1.03 | 0.05 | 0.17 | 0.71 | 0.31 | 1.46* |
| **FusionAD** | 0.81 | **0.02** | **0.08** | 0.27 | **0.12** | **0.37** |

to enhancing the prediction outcomes. In terms of planning results, a noticeable improvement is observed when fusion with a vectorized representation of past trajectories and ego status.

Table 5: Ablation studies for designs in the motion forecasting module.

| ID | Refine | Mode | minADE $\downarrow$ | minFDE $\downarrow$ | MR $\downarrow$ | minFDE-mAP $\uparrow$ | EPA $\uparrow$ |
|---|---|---|---|---|---|---|---|
| 1 | | | 0.394 | 0.636 | 0.088 | 0.507 | 0.622 |
| 2 | ✓ | | 0.395 | 0.627 | 0.086 | 0.516 | 0.624 |
| **3** | ✓ | ✓ | **0.388** | **0.617** | **0.086** | **0.516** | **0.626** |

Table 6: Ablation studies for designs in the planning module.

| ID | loss | ego status | $DE_{avg}$ | $CR_{1s}$ | $CR_{2s}$ | $CR_{3s}$ | $CR_{avg}$ | $CR_{traj}$ |
|---|---|---|---|---|---|---|---|---|
| 1 | | | 1.08 | 0.28 | 0.13 | 0.32 | 0.24 | 0.71 |
| 2 | ✓ | | 1.03 | 0.25 | 0.13 | 0.25 | 0.21 | 0.56 |
| 3 | ✓ | ✓ | **0.81** | **0.02** | **0.08** | **0.27** | **0.12** | **0.37** |

## 4.4 QUALITATIVE RESULTS

The comparative qualitative results between FusionAD and UniAD are depicted in Figure 5. The integration of Lidar sensory inputs and the novel design of the FMSPnP module in FusionAD demonstrates an enhancement in perception and prediction performance. For instance, Figure 5a illustrates a significant heading error in the bus detection by UniAD, attributable to the distortion from the camera, particularly in the overlapping region between the front and front-right cameras. In contrast, FusionAD accurately identifies the bus's heading. Figure 5b presents a prediction scenario involving a U-turn. FusionAD consistently predicts U-turn trajectories, whereas UniAD generates moving forward, left-turn, and U-turn modes. Please see our supplementary materials for video comparison.

While the proposed method demonstrates strong quantitative and qualitative performances, it still relies on a rule-based system for post-processing the output in order to achieve reliable real-world performance. Furthermore, the current research work (Hu et al., 2023b; 2022; Jiang et al., 2023) primarily evaluates the learned planner using open-loop results of planning tasks, which may not effectively gauge its performance. Evaluating the planner in a close-loop manner with real-world perception data poses challenges. Nonetheless, the prediction results under the end-to-end framework remain promising, and there is potential for further improvement of the planning module within this framework.

## 5 CONCLUSIONS

We propose FusionAD, a novel approach that leverages BEV fusion to facilitate multi-sensory, multi-task, end-to-end learning, thereby significantly enhancing prediction and planning tasks in the realm of autonomous driving. Instead of naively replacing the perception module to a fusion method, we design a uniformed prediction and planning module to integrate multi-modal features.

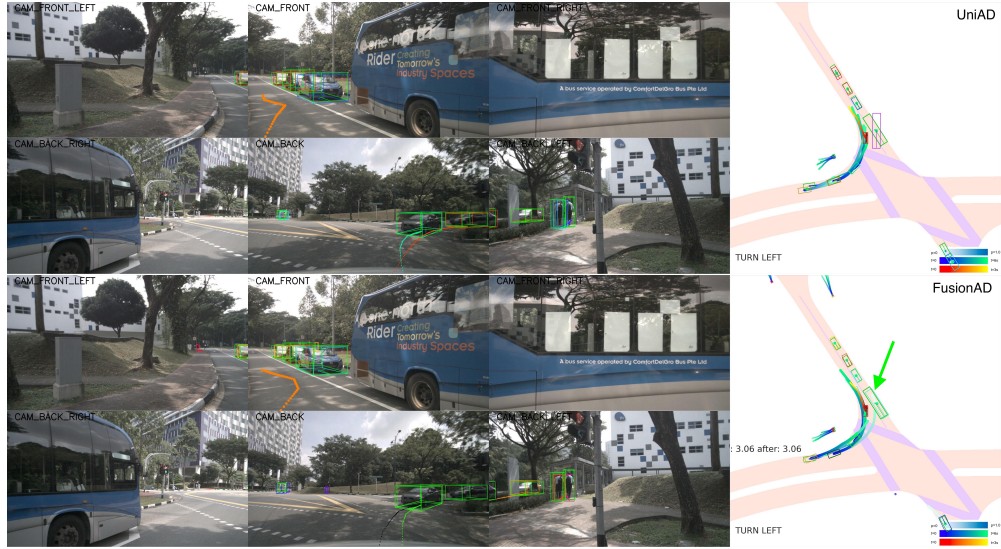

(a) Case 1: Perception of a bus. FusionAD detects the heading correctly while distortion exists in a near range, but UniAD incorrectly predicts the heading.

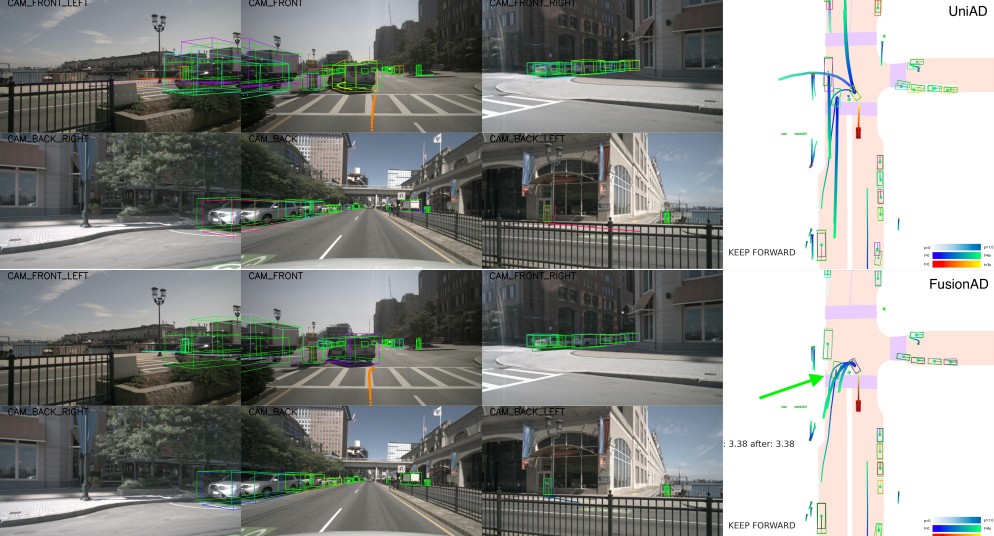

(b) Case 2: Prediction of U-turn. FusionAD consistently predicts the U-turn earlier in all modes which aligns with the ground-truth trace, while UniAD still predicts the move-forward, left-turn, and U-turn modes until the very last second U-turn actually happens.

Figure 5: **Visual comparison of two example cases between UniAD (Hu et al., 2023b) (Top) and our FusionAD (Bottom).**

The proposed approach has yielded substantial performance improvements in both prediction and planning tasks and has notably improved perception tasks when compared to end-to-end learning methods solely reliant on camera-based BEV. In future, we will focus on desiging efficient version of FusionAD in order to facilitate deployment on commodity vehicles.

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

## A  DETAILS OF BEV ENCODER

**LiDAR.** The raw LiDAR point cloud data is first voxelized and then used to generate LiDAR BEV features based on the SECOND network.

**Camera.** The multi-view camera images are first processed through a backbone network for feature extraction. Afterward, the FPN network is employed to generate multi-scale image features.

We further develop the following techniques to efficiently improve the performance of the fusion module.

**Points Cross-Attention.** During the points cross-attention process, each BEV query only interacts with the LiDAR BEV features around its corresponding reference points. This interaction is achieved using deformable attention:

$$\text{PCA}(Q_p, B_{\text{LiDAR}}) = \text{DefAttn}(Q_p, P, B_{\text{LiDAR}}) \tag{5}$$

where $Q_p$ represents the BEV query at point $p = (x, y)$, and $B_{\text{LiDAR}}$ represents the BEV feature output from the LiDAR branch. $P$ is the projection of the coordinate p=(x,y) in the BEV space onto the LiDAR BEV space.

**Image Cross-Attention.** To implement image cross-attention, we follow a similar approach to BEV-Former. Each BEV query is expanded with a height dimension similar to the Pillar representation. A fixed number of $N_{ref}$ 3D reference points are sampled in each pillar along its $Z$-axis. And the image cross-attention process is shown below:

$$\text{ICA}(Q_p, F) = \frac{1}{V_{hit}} \sum_{i=1}^{V_{hit}} \sum_{j=1}^{N_{ref}} \text{DefAttn}(Q_p, P(p, i, j), F_i) \tag{6}$$

where $V_{hit}$ denotes the number of camera views to which the reference point can be projected, $i$ is the index of the camera view, $F_i$ represents the image feature of the i-th camera, and $P(p, i, j)$ represents the projection of the 3D reference point $(x, y, z_i)$ of the BEV Query $Q_p$ onto the image coordinate system of the i-th camera.

**Temporal Self-Attention.** We also leverage the insights from BEVFormer to implement Temporal Self-Attention. Specifically, our approach involves temporal alignment of the historical frame BEV

features based on the vehicle's motion between frames. We then utilize Temporal Self-Attention to fuse historical frame BEV features, as shown below:

$$\text{TSA}(Q_p, (Q, B'_{t-1})) = \sum_{V \in \{Q, B'_{t-1}\}} \text{DefAttn}(Q_p, p, V) \tag{7}$$

where $B'_{t-1}$ represents the BEV features at timestamp $t-1$ after temporal alignment.

Since we are interested in the prediction and planning enhancement, for the detection, tracking, and mapping tasks in perception, we mainly follow the setting in UniAD (Hu et al., 2023b).

