# OpenReview forum: "FusionAD: Multi-modality Fusion for Prediction and Planning Tasks of Autonomous Driving"
_ICLR.cc/2024/Conference — ICLR 2024 Conference Withdrawn Submission_

### Official Review · Reviewer_Zqsu · 2023-10-28

**Soundness:** 3 good
**Presentation:** 3 good
**Contribution:** 2 fair
**Rating:** 5
**Confidence:** 5

**Summary:**

The author proposed a multi-modal fusion framework for joint perception, prediction, and planning tasks. The proposed framework leverages both multi-modal sensor data and temporal BEV features, achieving leading performance on nuScenes dataset.

**Strengths:**

* The proposed framework leverages both multi-modal sensor fusion and temporal fusion, exploring a new fusion paradigm for end-to-end autonomous driving.
* The paper presentation is good and is easy to follow.

**Weaknesses:**

* The technical novelty is unclear. The sensor fusion of BEV feature maps are mainly from FusionFormer as mentioned by the author. The modification for prediction and planning seems marginal compared with the existing UniAD. Fully discussions on the differences are needed.
* The proposed methods show superior performance than UniAD. What is the main contribution of this performance? Ablation study on perception, planning, and prediction pipeline with fixed modality input (e.g., only LiDAR, only camera, both modalities) is also needed to justify the performance boost and technical novelty. Right now, it is unclear if the performance gain is from the multi-modality or the pipeline design choice.
* What is the definition of used mADE, and mFDE? Trajectory prediction usually use mADE_k/mFDE_k to evaluate the accuracy of prediction with the consideration of multi-modality of multiple future paths. Please specify the k value if using mADE_k.

**Questions:**

Please see the weakness sections.

---

### Official Review · Reviewer_eLPm · 2023-10-31

**Soundness:** 3 good
**Presentation:** 2 fair
**Contribution:** 2 fair
**Rating:** 3
**Confidence:** 5

**Summary:**

This work presents a framework for integrating information from cameras and LiDAR for prediction and planning tasks. The architecture consists of a transformer-based multimodal fusion network followed by a fusion-aided modality-aware prediction and status-aware planning modules. Extensive experiments and ablations on the nuScenes dataset shows the benefits of multimodal fusion over UniAD and other baselines on motion forecasting and planning tasks.

**Strengths:**

- This work presents a multimodal multitask framework for prediction and planning tasks.
- The architecture consists of novel fusion-aided modality-aware prediction, status-aware planning modules and fusion-based collision loss modeling.
- Extensive experiments on nuScenes (Table 1) shows benefits on several tasks - detection, tracking, mapping, motion forecasting, occupancy precition, planning. Several baselines are considered (Table 2,3,4) and ablations (Table 5,6) show the effectiveness of different components.

**Weaknesses:**

- Existing transformer-based multimodal multitask end-to-end learned driving models [1,2] have shown the benefits of using camera and LiDAR information for downstream control. How does the proposed fusion module compare to the fusion modules in these works? The main difference seems to be that the camera features are explicitly transformed into the BEV space using BEVFormer-based encoder whereas in TransFuser, there is no explicit BEV transformation of image features. It'd be helpful to compare to TransFuser's fusion module to understand the differences and identify the benefits of the proposed module.
- The planning and motion forecasting experiments use open-loop evaluation settings. However, recent works [3,4] indicate that open-loop evaluation may not be representative of closed-loop performance. There already exists frameworks for closed-loop evaluation, eg. nuPlan[5] for planning and CARLA[6] for driving. It'd be useful to conduct experiments in these settings so that the results are more representative of the downstream control task.
- Minor comment: Why is training done in a 3-stage manner? Is it because of the high computational demand or would jointly training also work?

[1] Chitta et al, TransFuser: Imitation with Transformer-Based Sensor Fusion for Autonomous Driving. TPAMI 2022
[2] Shao et al, ReasonNet: End-to-End Driving With Temporal and Global Reasoning. CVPR 2023
[3] Codevilla et al, On offline evaluation of vision-based driving models. ECCV 2018
[4] Dauner et al, Parting with Misconceptions about Learning-based Vehicle Motion Planning. CoRL 2023
[5] Caesar et al, nuplan: A closed-loop ml-based planning benchmark for autonomous vehicles. CVPR Workshops 2021
[6] Dosovitskiy et al, CARLA: An open urban driving simulator. CoRL 2017

**Questions:**

I have 2 main concerns (see weaknesses for more details):
- How does the proposed fusion module compare to TransFuser line of work (since TransFuser also proposes a multimodal multitask driving framework for downstream control task)? The differences and benefits of the proposed fusion module over TransFuser are not clear.
- Recent works indicate that open-loop evaluation may not be representative of closed-loop performance. Since there are already closed-loop evaluation frameworks, eg. nuplan & CARLA, being used in driving tasks, it'd be useful to experiment in this setting for reliable results.

---

### Official Review · Reviewer_w8af · 2023-11-01

**Soundness:** 3 good
**Presentation:** 3 good
**Contribution:** 3 good
**Rating:** 6
**Confidence:** 4

**Summary:**

This work introduces FusionAD, a multi-modality multi-task model for autonomous driving.
FusionAD is a single model that performs a variety of perception tasks (detection, tracking, map prediction, occupancy) and planning oriented tasks (trajectory prediction, motion planning) from raw lidar and camera inputs.
The model follows a transformer architecture and they devise a fusion module (FMSPnP) to share tokens between different high level tasks (prediction and planning).

**Strengths:**

* Quantitative evaluation looks strong compared to the presented baselines. Results cover a wide variety of tasks on the nuScenes dataset.
* Paper is well written and easy to follow. Figure are good quality and informative.
* Putting together a unified model is a solid contribution and requires a lot of careful engineering behind the scenes - it would be nice if the authors could share a list of findings from their development process.

**Weaknesses:**

* Comparisons with prior work - the numbers are convincing across all different tasks (Table 1-4), but my first thought was that it comes from the fact that FusionAD is using lidar inputs. For all baselines it would be great to have an extra column that denotes which input modalities are used.
* Camera+lidar fusion has proven to be effective across perception tasks, so it seems that the main idea the paper pitches is the need for interactions between the high level modules (perception/planning). If this is the case, the most simplest baseline to discuss or compare with is UniAD with additional lidar input.
* Ablations (Table 5, 6) are not that informative to overall idea.

**Questions:**

Have the authors performed any profiling? I am curious to see the inference time/performance compared to each of the corresponding state-of-the-art in the set of tasks.

It would be nice to have a detailed description of each of the tasks in the supplementary to clarify any missing details - i.e. occupancy prediction could refer to BEV occupancy or 3D occupancy.

---

### Official Review · Reviewer_ynwV · 2023-11-02

**Soundness:** 2 fair
**Presentation:** 2 fair
**Contribution:** 2 fair
**Rating:** 3
**Confidence:** 4

**Summary:**

This paper presents FusionAD, a hierarchical multi-task autonomous driving network in the style of recent work UniAD for tracking, mapping, motion prediction, occupancy and open-loop planning. The UniAD model is extended in 3 ways: (1) an updated backbone network which is based on FusionFormer (camera + LiDAR + time) instead of BEVFormer (camera + time); (2) a modified prediction module with a multi-stage predict-and-refine formulation; and (3) a modified planning module with a new collision loss during training and ego status input. Besides this, the architecture is trained in a new 3-stage process. This leads to an approach with strong empirical performance across all tasks on the nuScenes dataset.

**Strengths:**

The key strengths of the paper are the strong empirical results of the final model and standardized experimental setup with established data and metrics, significantly pushing the state-of-the-art. The results are well-structured, demonstrating improvements both quantitatively and qualitatively. Besides this, the proposed modifications are simple and well-motivated and the paper is clearly written.

**Weaknesses:**

1. Inconsistent depiction of novelty over UniAD in Fig. 1b: UniAD has camera-only features, but an identical hierarchical structure for perception, prediction and planning as the proposed FusionAD from Fig. 1c. The current figure creates the impression of a new contribution with respect to the downstream architecture, while in the paper the only contributions are towards existing modules in the UniAD framework (backbone, forecasting, planning).
2. Table 6 shows that gains in open-loop planning performance over UniAD result from only the new ego-status input. Recent work has already shown that including ego-status as a planner input significantly simplifies the task [1,2], leading to SoTA results on nuScenes even with ego-status as the only input [1]. The discussion should clearly mention that the new sensor fusion backbone does not contribute to open-loop planning performance.
3. Unclear significance of proposed changes to MotionFormer for prediction: Table 5 shows a very minor impact towards motion forecasting performance with the proposed changes (0.394→0.388 MinADE), which means that nearly all gains in performance with respect to UniAD (0.708→0.388 MinADE) come from the FusionFormer based backbone.
4. Missing related work: the idea of multi-modal fusion in multi-task end-to-end autonomous driving stacks is well-established in all the top entries of the CARLA simulator [3,4,5,6,7] but these are not discussed in the paper. In Section 2.3, it is incorrectly claimed that TransFuser does not use BEV space, even though the BEV branch is a key component of the TransFuser architecture.

[1] https://arxiv.org/abs/2303.12077

[2] https://arxiv.org/abs/2305.10430

[3] https://arxiv.org/abs/2203.11934

[4] https://arxiv.org/abs/2205.15997

[5] https://arxiv.org/abs/2207.14024

[6] https://arxiv.org/abs/2305.10507

[7] https://arxiv.org/abs/2306.07957

**Questions:**

1. Please see “Weaknesses” - these are the key points with the most influence on my rating. Taken together, W2 and W3 show that the proposed changes to the prediction and planning modules provide limited insights.
2. Unclear separation of methods in Table 4: different methods in Table 4 have different input modalities (camera, LiDAR, ego status). Please add columns clearly showing this difference.
3. Would it be possible to provide a comparison to UniAD in terms of model size, training time and inference time? This is valuable information from a practitioner’s perspective.
4. I would recommend making the differentiation between UniAD and FusionAD clearer in Fig. 3 and Fig. 4 by highlighting the new architectural components.